# Stochastic Computing Emulation of Memristor Cellular Nonlinear Networks

**DOI:** 10.3390/mi13010067

**Published:** 2021-12-31

**Authors:** Oscar Camps, Mohamad Moner Al Chawa, Stavros G. Stavrinides, Rodrigo Picos

**Affiliations:** 1Industrial Engineering and Construction Department, University of Balearic Islands, 07122 Palma Mallorca, Spain; oscar.camps@uib.es; 2Institute of Circuits and Systems, Technical University of Dresden, 01062 Dresden, Germany; mohamad_moner.al_chawa@tu-dresden.de; 3School of Science and Technology, International Hellenic University, 57006 Thessaloniki, Greece; s.stavrinides@ihu.edu.gr; 4Health Institute of the Balearic Islands (IDISBA), 07120 Palma Mallorca, Spain

**Keywords:** cellular nonlinear networks, stochastic logic, real time processing, image processing, memristors

## Abstract

Cellular Nonlinear Networks (CNN) are a concept introduced in 1988 by Leon Chua and Lin Yang as a bio-inspired architecture capable of massively parallel computation. Since then, CNN have been enhanced by incorporating designs that incorporate memristors to profit from their processing and memory capabilities. In addition, Stochastic Computing (SC) can be used to optimize the quantity of required processing elements; thus it provides a lightweight approximate computing framework, quite accurate and effective, however. In this work, we propose utilization of SC in designing and implementing a memristor-based CNN. As a proof of the proposed concept, an example of application is presented. This application combines Matlab and a FPGA in order to create the CNN. The implemented CNN was then used to perform three different real-time applications on a 512 × 512 gray-scale and a 768 × 512 color image: storage of the image, edge detection, and image sharpening. It has to be pointed out that the same CNN was used for the three different tasks, with the sole change of some programmable parameters. Results show an excellent capability with significant accompanying advantages, such as the low number of needed elements further allowing for a low cost FPGA-based system implementation, something confirming the system’s capacity for real time operation.

## 1. Introduction

Cellular Nonlinear Networks (CNN) were introduced by Chua and Yang [1] in 1988, and can be described as a mixture between Cellular Networks and Artificial Neural Networks that can implement a parallel processing universal computer machine. This bio-inspired architecture is able to process in parallel massive amounts of data, thus being specially suitable for image processing, with single ASIC CMOS prototypes already implemented being able to deal with rates up to $3\times 10^{4}$ frames per second [2].

Memristors have been proposed as a device that may help to implement this kind of circuit [3,4], but experimental implementations are still lacking. These devices, memristors, are passive, two-pole elements also introduced by Chua in 1971 [5], as a theoretically possible basic circuit element. In 2008, Strukov et al. [6] realized their ReRAM devices were, actually, a kind of memristor. There have been many groups dedicated to creating either devices or emulators, ever since. One of the more classical mathematical memristor descriptions including memconductance *G* can be written as:(1)$$\begin{array}{c} i(t)=G(Q)\cdot v(t) \end{array}$$
where *Q* (also known as charge) is the integral over time of the current *i*: (2)$$\begin{array}{c} Q\left(t\right)=\int^{t}i(t)dt \end{array}$$

Notice that the requirement for the device to be a memristor is mapped to the requirement for the characteristics of the device to be dependent on some internal variables, as will be further discussed below, in addition to some fingerprints [7,8].

One of the main problems for using memristors into circuits is that they are not yet readily available for implementation in usual technologies. In this paper we use Stochastic Computing to implement a fully digital realization of a CNN using memristors. To do so, we have used a memristor emulator presented in [9], as well as a Stochastic Computing implementation of a CNN using it, as in our previous work [10], where a simpler implementation was presented operating exclusively on gray images. The notion of Stochastic Computing (SC) was introduced by Von Neumann in the 1950s [11] as a theoretical framework to explain how (relatively) accurate results could be obtained using imprecise systems. In this framework, time and accuracy are balanced in a trade-off. It was later popularized by Gaines [12], and it has found a niche in approximate computing. There are many examples in the literature for diverse applications, such as data compression [13], chaotic equation calculation [14], data mining [15], FFT computation [16], control [17], image processing algorithms [18,19], or A/D conversion [20]. As of today, it seems probable that it may conquer an important share in edge computing applications, since it can decrease energy consumption and circuit complexity and area overhead for low numbers of bits. However, this comes at a price, since the time required to perform the operation also increases exponentially with the number of bits. This, in turn, may increase the total energy consumption when the number of bits exceeds 16–17 [21,22]. There are, however, techniques that allow this problem to be alleviated, and make this competitive even for higher bit-numbers [22].

Within this paradigm, strings of $2^{N}$ binary digits $b_{i}\in \left[0,1\right]$ are used to represent real numbers *p*. These strings of digits are called stochastic computing numbers (SCN) and their mean value corresponds to the number *p* represented [23]. Thus, the absolute value ranges for any implementation of SCN must fall within [0,1]. There are two distinct possibilities to map a real number to SCN: the first one is from the domain [0⋯1], and the second one corresponds to [−1⋯1].

Once the selected mapping has been chosen, the different SCN operations can be easily implemented, requiring only the use of fundamental logic gates or very simple digital circuits. As an example, in the case of the [0⋯1] domain, multiplication of SCNs are implemented using an AND gate; in the second case, the [−1⋯1] domain, this same operation demands using a XNOR gate.

The full system we have implemented is depicted in Figure 1. The first part is processed in the computer, where the images are read using a Matlab script and converted to gray scale if needed. The resulting image is then sent to the FPGA board, which is used as an accelerator and connected using the FPGA-in-the-loop methodology that allows to integrate it in a seamless way. The FPGA performs the processing using a massively parallel implementation of the proposed Stochastic Memristive Cellular Nonlinear Network, and the result is then read back into the computer and represented to the user, along with the entropy of the image and the rms error if needed.

The paper is structured as follows: after this introduction, the basics of memristors and Cellular Nonlinear Networks are presented in Section 2, which is being used in Section 3 to implement the basic CNN cell in Stochastic Computing. Section 4 presents the results obtained using three different pictures (two gray, one color) with three different sets of parameters. These three sets of parameters allow the CNN to perform three different operations on the images: storing, edge detection, and image improvement. Finally, Section 5 concludes the paper.

## 2. Memristive Cellular Nonlinear Networks

### 2.1. Memristors and Memristive Systems Modeling

Among the possible theoretical descriptions of memristive systems, Corinto et al. present in [24] a very complete framework to study memristors and, in general, systems that may present memristive behavior. They propose using both the classical description using voltage and current, but they also discuss the flux–charge (φ–*q*) approach.

The memristive systems can be classified according to how far they are from ideality. Following the taxonomy proposed in [7], there are three distinct possibilities: the ideal, the generic, and the extended memristor. This extended categorization was a theory requirement, needed to cover the description of pinched, hysteretic behaviors found in numerous new various elements.

The most general class of memristors are the extended memristors. The memristors belonging to this class are described by extra internal state variables (in addition either to current and voltage, or to φ and *q*). As an example, Equations (Equation 3) to (Equation 5) implement the case of flux-controlled memristors:(3)$$\begin{array}{c} i=G(\varphi ,v,x)\cdot v \end{array}$$
(4)$$\begin{array}{c} \dot{x}=g_{\varphi }\left(\varphi ,v,x\right) \end{array}$$
(5)$$\begin{array}{c} \dot{\varphi }=v \end{array}$$

This way, an extended memristor has a memristance *M* represented by the nonlinear memconductance *G* (or, more accurately, its inverse) in Equation (Equation 3), where φ is the flux, and *v* is the voltage between the terminals of the memristive device. The extra variables are grouped into the vector x, and they may comprise different physical magnitudes depending on the specific memristive system; as examples, we can mention the radius of a conducting filament, the internal temperature of the system, as well as other non-electrical variables that may be used to describe the state of the memristor. These state variables x present a dynamic behavior described by $g_{\varphi }$ and Equation (Equation 4). As a side comment, it is worth noticing that all the devices described as being memristors are indeed extended memristors.

When no parasitic effects are present, those extended memristors are better described as generic memristors (or, simply, memristors), since function $g_{\varphi }$ depends only on the state variables x and φ. Ideal memristors, finally, are those corresponding to the original definition [5], and can be considered in this framework as generic memristors with no state variable dependence other than charge or flux.

As an example, let us consider the simplest memristor model that can be devised, similar to those discussed in [24] or, for real devices, in [25,26,27,28]. In this case, the model of an ideal memristor where the memresistance or the memconductance depends only on the charge or the flux can be written as: (6)$$G=G_{0}\left(1+\frac{\phi }{\phi_{0}}\right)$$
where $G_{0}$ is the unperturbed conductance, and $\phi_{0}$ includes the importance of the memristive effect. Notice that for $\phi_{0}\to \infty$, the behavior tends to be similar to that of an ideal resistor. The behavior of the device is represented in Figure 2, for $G_{0}=0.1\;mS$, $\phi_{0}=10$, and three different frequencies. Notice how the device reproduces the two fingerprints of a memristor [7,8]: it presents a pinched loop whose area tends to zero at high frequency (green line, ‘x’ symbol).

### 2.2. Cellular Nonlinear Networks

The system discussed in this paper, Cellular Nonlinear Network (CNNs) [1], is not to be confounded with Convolutional Neural Networks (also CNN), even if they share the same acronym. The CNNs discussed here represent a powerful massively parallel, multivariate signal processing paradigm. In their most basic description, they are made of independent processing units, called *cells*, where each cell has an input, an output which is fed back as another input, and also feels the effect of the inputs and outputs of its nearest neighbors. These effects are then processed internally into a state variable, and the output is linearly dependent on the result of this processing, with a positive and negative saturation.

As an example of hardware implementation of a CNN, we find [2], where each processing element typically accommodates additional data storage units, which allow the CNNs to store the programming parameters at the cell level. As a result, these Universal Machines (UMs) can be considered as one of the earliest examples of a non-von Neumann computer. Unfortunately, these memory blocks need a large integrated circuit (IC) area to be implemented, which increases significantly the size of each cell. As a consequence, the spatial resolution is quite poor compared with simple image sensors, which is a common problem that CNN-UMs and, as an inherited problem, arrays comprising sensor-processor cells based upon them, suffer from.

Mathematically, the behavior of the i,j cell can be described by a differential equation as:(7)$$\begin{array}{cc} \frac{dx_{ij}}{dt}= & -x_{ij}+a_{0,0}f\left(x_{ij}\right)+z_{ij}+b_{00}u_{ij}\\ \; & +\sum_{k,i\in N_{i,j},k\neq i,l\neq j}a_{k-i,l-j}y_{kl}\\ \; & +\sum_{k,i\in N_{i,j},k\neq i,l\neq j}b_{k-i,l-j}u_{kl} \end{array}$$
where $x_{ij}$ is the state variable, $u_{ij}$ and $y_{ij}$ are the inputs and outputs, respectively, and $a_{ij}$ and $b_{ij}$ are the *feedback* and *feed-forward* coefficients. The stability of the system can be controlled by setting the value $z_{ij}$ to an appropriate value [4], and is equivalent to a constant bias in the electrical equivalent. Function f(x) is a nonlinear function, saturating to a minimum and a maximum values ($v_{min}$ and $v_{max}$). In this sense, it is similar to the output function of a neuron. However, the most usual shape for it is a piecewise linear function, defined as:(8)$$f\left(v\right)=\frac{1}{2}\left(\left|v+v_{sat}\right|-\left|v-v_{sat}\right|\right)$$

Equation (Equation 7) is usually rearranged as:(9)$$\begin{array}{cc} \frac{dx_{ij}}{dt}= & g\left(x_{ij}\right)+z_{ij}+b_{00}u_{ij}\\ \; & +\sum_{k,l\in N_{i,j},k\neq i,l\neq j}\left(a_{k-i,l-j}y_{kl}+b_{k-i,l-j}u_{kl}\right) \end{array}$$
(10)$$g\left(x_{ij}\right)=-x_{ij}+a_{0,0}f\left(x_{ij}\right)$$

Notice that, even if the sum is made over the whole set of integers, we usually restrict ourselves to just the nearest neighbors. This Equation (Equation 10) can be implemented with a single element thanks to the unique nonlinear behavior of memristors. Thanks to this capability to process or store data within a common physical nanoscale medium, their use in future CNN cell designs may allow to remove the burden of extra memory blocks within each processing element, allowing the development of co-located sensor-processor arrays with enormous high-resolution levels, specially suited for the Internet of Things (IoT) industry.

Referring to the determination of the coefficients, some methods have been published [29,30] for the traditional CNN, and, more recently, using the so-called Dynamic Route Map (DRM) [3,4,31,32], to the first-order approximate model of each cell of a Memristor CNN (M-CNN). Such an approximation is depicted as a circuit in Figure 3, and the evolution of the state variable ($x_{ij}$) is described by an equation equivalent to Equation (Equation 9):(11)dxijdt=k(xij)[ +θ(vx;i,jf+p(xm;i,j)+θ(−vx;i,jf−p(xm;i,j)]
where *i* ∈ 1,..., M, *j*∈ 1,..., N, θ() is the unit step function, $v_{x_{i,j}}$ stands for the voltage across the capacitor $C_{x}$, whereas $xm_{i,j}$, and $v_{m_{i,j}}\equiv v_{x_{i,j}}$ denote, respectively, the state and voltage of memristor $m_{x}$, whose current is described via the generalized Ohm’s law from Equation (Equation 3): $i_{m_{i,j}}=G\left(x_{m_{i,j}}\right)v_{m_{i,j}}$, with the memductance given by $G\left(x_{m_{i,j}}\right)=x_{m_{i,j}}^{-1}$. The non-linear function proposed in [32] to characterize the memristive CNNs are:(12)$$k\left(v_{x_{i,j}}\right)=-\beta v_{x_{i,j}}+\frac{\beta -\alpha }{2}\left(\left|v_{x_{i,j}}+V_{t}\right|-\left|v_{x_{i,j}}-V_{t}\right|\right)$$
with $\beta >\alpha \in {ℜ}^{+}$, featuring units $\Omega V^{-1}s^{-1}$, and $V_{t}\in {ℜ}^{+}$ denoting the minimum voltage needed by the memristor for switching. In addition, there are two different window functions, to ensure that the memristor stays in between the two possible states $\left[x_{on},x_{off}\right]$. These two window functions can be written in a compact way [32] as:(13)$$f_{r}^{p}\left(x\right)=1-{\left(\xi +\frac{x-xon}{x_{off}-xon}\right)}^{2p}$$
where ξ=−1 when r= “+” (the upper boundary), and ξ=0 in the opposite case; *p* can be any integer (p∈Z). In addition, the dynamical evolution of the voltage $v_{x_{i,j}}$ of each cell in system presented in Figure 3 is:(14)$$\begin{array}{cc} C_{x}\frac{dv_{x_{i,j}}}{dt} & =-\left(\frac{1}{R}+G\left(x_{m_{i,j}}\right)\right)\cdot v_{x_{i,j}}+a_{0,0}f_{out}\left(v_{x_{i,j}}\right)\\ \; & +i_{bias}+b_{0,0}v_{in_{j,k}}\\ & +\sum_{k,l\in [-1,1]}\left(b_{k,l}v_{in_{i+k,j+l}}+a_{k,l}v_{out_{i+k,k+l}}\right) \end{array}$$
(15)$$f_{out}\left(v\right)=\frac{R_{y}g_{lin}}{2}\left(\left|v+v_{sat}\right|-\left|v-v_{sat}\right|\right)$$

Notice the equivalence of Equation (Equation 14) with Equation (Equation 9): the last line of both equations is equivalent, while the first line of Equation (Equation 9) corresponds to the first two lines of Equation (Equation 14). Thus, we can conclude that the circuit in Figure 3 accurately represents a possible implementation of a CNN cell, and it is the circuit we will implement in the next section as a SC module.

## 3. M-CNN Stochastic Computing

### 3.1. Stochastic Computing Basics

The SC approach adopts real number representation by strings of *N* random binary numbers $b_{i}$. The probability of “1” bits to appear within the bit-string is proportional to the number to be operated [23]:(16)$$p=\frac{1}{N}\sum_{i}b_{i}$$

These strings are called Stochastic Computing Numbers (SCN) or Stochastic Encoded Numbers (SEN). In this paper, we will opt for the second, using also the term Binary Encoded Numbers (BEN) for those encoded as classical binary numbers. Notice that the number of bits that can be encoded in a chain of length *N* is $log_{2}N$, since the relevant information is just the number of “1”s. There are two main ways to generate a map between a SCN and real numbers: first, we can map the desired range of real numbers to the real domain [0⋯1]; second, we can map them to the interval [−1⋯1]. Depending on which mapping is to be implemented, many different mathematical operations can then be performed using simple logic gates or simple sequential circuits.

As an example, multiplication of SCN is performed using a simple AND gate when using the [0⋯1] domain. Alternatively, considering the [−1⋯1] domain, the same multiplicative operation requires the use of an XNOR gate, as shown in Figure 4a.

Since we cannot represent any SC number as a probability higher than one, the case of addition becomes slightly more complex, since 1+1=2. Thus, the operation that should be implemented is (x+y)/2, which would always return a maximum value of 1. This operation is usually implemented using a multiplexer, as shown in Figure 4b, where the *p*(0.5) indicates a signal with a probability of 50% of being ‘1’ or ‘0’. This necessary input signal is generated using one of the bits generated in the RNG, so no additional circuitry is needed. It is worth pointing out that this gate is the same in both the [0⋯1] and the [−1⋯1] domains. Other more complex operations (division [17], square roots [17], reversible gates [33], etc., ⋯) are also discussed in the literature, although not presented in this paper.

Another important point is the conversion from BEN to SEN. This is usually achieved by using a scheme similar to that in Figure 5, where an N-bit random number is generated by utilizing a random number generator (RNG) and compared to the value of the N-bit BEN. If the output of the RNG is below the BEN, the converter’s output would be bit “1”, and bit “0” otherwise. In the opposite operation, converting SEN back to its BEN representation, the number of 1’s included in the signal needs to be calculated; something that can be straightforwardly achieved by a simple counter.

It is apparent that the error in the approximation of the SEN to its actual value is equivalent to the error provided by a random walk process of length *n*, and thus proportional to $\sqrt{n}$, as it has been discussed in the literature [34]. Therefore, using *N* bits, we may consider that all the noise caused by the process is included in the lowest N/2 bits. This way, the noise figure NF for a signal of power $S_{p}$ with noise power $N_{p}$ caused by the use of the SEN is:(17)$$NF=10log_{10}\left(\frac{S_{p}}{N_{p}}\right)=10log_{10}\left(\frac{2^{N}}{2^{N/2}}\right)\approx 3.01\;N/2\;\mathrm{dB}$$

Notice that for this equation we have considered the maximum possible amplitude for the input signal. In order to consider the minimum amplitude over the noise, we consider we are 1 bit over the noise (N/2+1). In this case, we obtain:(18)$$NF=10log_{10}\left(\frac{S_{p}}{N_{p}}\right)=10log_{10}\left(\frac{2^{N/2+1}}{2^{N/2}}\right)\approx 3.01\;\mathrm{dB}$$

Thus, the system is expected to have a NF between 3 dB and $3\sqrt{N}$ dB. This desired NF would set the required number of bits, which is related to the sensitivity of the equation system, on noise. Empirically, we have seen that linear equations allow for a low *N*, while nonlinear systems call for higher values. Notice that a value of the NF=20 dB calls for N=12 bits, while for N=32 bits the provided noise figure NF=54 dB.

### 3.2. Stochastic Computing Implementation of a Memristor Emulator

The described advantages of SC framework were used by a stochastic computing implementation of a memristor emulator [9], which describes the memristor using Equation (Equation 6), and was written in the form of Equations (Equation 19)–(), so that it could be implemented in a discrete way. The mathematical operations Equations (Equation 19)–() were carried out by simple digital gates. A simple way to model memristors is using a linear relation of the charge *Q* with the memconductance G(Q), with an upper and a lower value, $G_{max}$ and $G_{min}$, correspondingly. Then according to this approach: (19)$$\begin{array}{c} G_{1}\left(Q\right)=G_{0}+G_{1}\cdot Q \end{array}$$(20)$$\begin{array}{c} G_{2}\left(Q\right)=min\left(G_{max},G_{1}\left(Q\right)\right) \end{array}$$(21)$$\begin{array}{c} G\left(Q\right)=max(G_{min},G_{2}\left(Q\right)) \end{array}$$

It is apparent that now Stochastic Computing can be used to implement such a model within a digital environment (an FPGA, or an ASIC). Other complex, physically based models simulating memristive behavior in FPGAs can be found in the literature [35,36], but they are very mathematically complex models, requiring a very large number of gates.

The emulator discussed above presents all the standard fingerprints of a memristor as required by the theory [8]. This implementation appears as a block diagram in Figure 6, where $v_{x}$ and GND are the SCN values of the positive and negative terminals of the memristor, respectively, while the calculated current is represented by the stochastic value $i_{M}$. The SCN value of GND has to be represented by a probability of 0.5 for an “1”, since we are mapping the interval [0⋯1] to an interval that includes negative and positive values. In our case, we used a public version of the Mersenne twister algorithm, available from GitHub [37].

Rewriting Equations (Equation 19)–() to allow them to be implemented in discrete time results in:(22)$$Q=\int^{t}i(t)\approx \Delta t\cdot \sum_{j}i(t=j\cdot \Delta t)$$
where the integration step is Δt. Using Equation (Equation 22), we can rewrite (Equation 19) as:(23)$$\begin{array}{c} G_{1}\left(Q\right)=G_{0}+G_{1}\cdot \Delta t\cdot \sum_{j}i(t=j\cdot \Delta t)S \end{array}$$

The adder needs to increase or decrease its output by a unit, depending on the inputs, since *i* and GND are both SEN. An increasing is performed when $v_{x}=1$, while when GND=1 the output is reduced. A single constant is used to group $M_{1}$ and Δt, and the max and min functions are built into the adder by establishing a maximum and a minimum values. A SEN output is generated from the adder’s output by comparing it to a random number spanning $\left[0\cdots \left(2^{NB}-1\right)\right]$. An AND gate is then used to obtain the current as per Equation (Equation 1), as in Figure 6.

### 3.3. Stochastic Computing Implementation of a M-CNN

As discussed above, a very compact implementation of a single cell of a Cellular Nonlinear Network can be performed as in Figure 3 [3]. The output voltage $v_{y;i,j}$ presents a non-linear dependence on the internal voltage $v_{x;i,j}$. The dynamic behavior of this $v_{x}$ is governed by a differential equation:(24)$$\frac{dv_{x}}{dt}=\frac{1}{C}\left(\sum_{i,j}\left(B_{i,j}i_{i,j}+A_{i,j}o_{i,j}\right)-i_{M}\right)$$
where $i_{i}$ are the input currents caused by the inputs of the nearest cells. The currents $o_{i}$ correspond to the outputs of those cells, while the current $i_{M}$ is that flowing through the memristor. As proposed in [3,4], we only consider the 8 closest neighbors. This way, Stochastic Computing can be used to make an implementation of this equation. As an initial step, we perform a first-order integration of (Equation 24):(25)$$\Delta v_{x}=\frac{\Delta t}{C}\left(\sum_{i}\left(B_{i}i_{i}+A_{i}o_{i}\right)-i_{M}\right)$$

The above Equation (Equation 25) has been implemented in Figure 7 as a stochastic equivalent circuit, where all data circulation correspond to 1-bit lines. Thus, implementation of Equation (Equation 25) was performed with 16 adders each implemented using 1 OR gate with a 1-bit multiplexers, 2 multipliers implemented as AND gates, 1 inverter, and 1 accumulator. Additionally, a random number generator is also needed (note that it may be shared between different cells) along with the memristor emulator previously presented.

The speed of the system can be estimated with the length of the chain needed to implement the SCN representation of the numbers to be used. These numbers are determined considering the input images. In our case, we are using both gray images and color images, all of them downloaded from http://www.hpca.ual.es/~vruiz/images (accessed on 20 June 2021). The gray images use a 8-bit single plane to store it, while the color images use three different 8-bit planes. Thus, at least 8 bits need to be recovered faithfully. As discussed above [34], the use of 14 bits corresponds to a chain length of $2^{14}=$ 16,384 bits and can represent values with an error confined in the last 6 bits, with more than a 95% probability. Following this reasoning, we choose these 14 bits for both the length of the chain and the accumulators.

## 4. Image Processing Results

The CNN described above was simulated using Matlab, with the circuit parameters equivalent to those appearing in Table 1. Parameter values have been chosen to be similar to those proposed in [9], to optimize the emulator behavior. Another option would have been using the process described in [3,32] or [38], but then scaling of the values was differing significantly to allow the emulator performing efficiently. The FPGA-in-the-loop methodology, as shown in Figure 1, was implemented to speed up the simulation, using an Arria V development kit. This FPGA system was connected to the computer running the Matlab code via a cabled network.

The *A* and *B* matrices were changed to three different sets, corresponding to three different cases: store, edge, sharpening, as discussed below. The notation to represent the coefficients in the matrices is shown in Table 2.

The images all have different sizes (notated as N×M), as reported in Table 3, and they were all initially color images with 8-bit color resolution at each RGB plane. We have processed these figures by performing two different experiments, both of them using three different (A,B) sets of parameters: first, we used a color-to-gray conversion, and we processed the images through the three different CNN. In a second step, we used a color image, and we processed each color plane independently to finally reconstruct a color image from these three planes.

In order to keep a good NF during the stochastic processing, and according to Equation (Equation 17) and the discussion in the previous section, each pixel was converted from 8 to 14 bits by padding the least significant positions with zeros. After the processing, the stochastic images to normal images were converted back by disregarding the 6 least significant bits of the corresponding accumulator in Figure 6.

The results are shown in two different ways: as pictures, and also using the RMS error and the entropy of the image. The RMS error $e_{rms}$ is defined in Equation (Equation 26), where $p_{i;i,j}$ and $p_{o;i,j}$ are the values of the pixel (i,j) for the input and the processed image, respectively. The entropy *H* is calculated using the Matlab implementation of Equation (Equation 27), where $p_{i}$ contains the normalized histogram counts for each gray level. The entropy of the unprocessed images is reported in Table 4. Notice that for the color images we report the values of the entropy for each channel, while for the rest we report only the entropy of the gray image.
(26)$$\begin{array}{c} e_{rms}=\frac{1}{N\cdot M}\sum_{i,j}{\left(p_{o;i,j}-p_{p;i,j}\right)}^{2} \end{array}$$
(27)$$H=-\sum_{i}p_{i}log_{2}\left(p_{i}\right)$$

### 4.1. Store Image

As a first example, we show the results of storing the image into the CNN. That is, the values of internal voltage are evolved until the output is equal to the input. As a comment, this is the easiest “program” that can be implemented into the SM-CNN, and can be used as a first step for more complex algorithms as can be, for instance, a background removal or a motion detection algorithms. The coefficients of the matrices are provided in Table 5.

We have represented both the input and output images for two different cases input gray images namely Figure 8a and Figure 9a (a zoom-in of the latter, appears in Figure 10a) and a color image (Figure 11a). The results for the store process are shown in Figure 8b and Figure 9b for the gray images, while the stored color image is presented in Figure 11b. Visually, it can be seen there that the algorithm performs correctly.

Additionally, we have calculated the entropy of the images and the RMS error, as shown in Table 4 and Table 6. The entropy of the images is nearly the same, and the rms error is kept, at most, below 1.3%.

### 4.2. EDGE Detection Using SM-CNN

The proposed SM-CNN was further checked using an implementation of one of the stochastic systems proposed in [4]. Specifically, we have improved the EDGE routine presented in [10]. This routine performs a border detection algorithm in the image using the coefficients in Table 7. In the previous work, the routine was fixed, and no quantitative analysis was performed. The edge algorithm aims to detect changes between adjacent pixels, so the output value will evolve to a 1 or 0, depending on the change of color. The evolution will depend on the threshold of the output function and can thus be changed.

We used the same images as in the previous example, where two of the images were gray 8-bit images and another one was a 24-bit color image (3×8 bits planes). They were processed as in the previous case to 14 bits by padding, and back to 8 bits by truncation. We have represented both the input and output images for two different gray images in Figure 8c and Figure 9c, while the result for the color image is shown in Figure 11c. It can be seen there that the algorithm performs as expected, showing also the corresponding decreasing in the values of the entropy in Table 4.

### 4.3. Sharpening

The sharpening algorithm is a variation of the EDGE detection, combined with the STORE genii. In fact, we calculated a new coefficient set as a linear combination of the two previous sets: $A_{s},B_{s}$ for the store matrices, and $A_{e},B_{e}$ for the edge detection. The new set $A_{i},B_{i}$ is calculated as:(28)$$M_{i}=\lambda_{1}M_{s}+\lambda_{2}M_{i}$$
where *M* stands for both *A* and *B*. In this case, $\lambda_{1}=1/3$ and $\lambda_{2}=2/5$, and the corresponding matrix coefficients are provided in Table 8.

We used the same images as in the previous example, where two of the images were gray 8-bit images and another one was a 24-bit color image (3x8 bits planes). They were processed as in the previous case to 14 bits by padding, and back to 8 bits by truncation. Results of applying this set of coefficients to gray images are shown in Figure 8d and Figure 9d (with a zoom comparing original in Figure 10a against the processed output in Figure 10b). The results for a color image are depicted in Figure 11d, with a zoom-in shown in Figure 12a,b for the original and processed images, respectively. Visually, the images seem to be improved, with less fuzzy edges. This is further corroborated by the increase in the entropy shown in Table 4.

## 5. Conclusions

In this work we have performed a fully digital implementation of a Memristive Cellular Nonlinear Network profiting from the Stochastic Computing paradigm. The basic unitary cell of the proposed CNN features a digital memristor emulator, plus several arithmetic units that are implemented as very simple gates, allowing for an enormous number of cells in parallel, which can translate into a very fast image processor.

We have implemented this full CNN structure into an ARRIA V FPGA, and we have tested it along with Matlab by implementing three different procedures: a image store, an edge detection, and, finally, an image sharpening process. Notice that these three procedures involve only the change of the matrix coefficients, that are common to all the cells. As has been discussed, the results imply that the system performs smoothly, with errors lower than 1.3% in the storage, an excellent edge detection capability, and a very good detail sharpening. A full FPGA implementation of images with lower number of pixels would allow for a very high image processing speed, adequate for real time needs, and well inside the capabilities and requirements of edge computing. In addition, as shown in [38], the proposed kind of memristive CNNs are resilient to individual “pixel” failures.

As an example, in this paper we have used 14-bit stochastic numbers, which translates into a chain length of $2^{14}=$ 16,384. An entry-level FPGA can run at 80MHz, so it could operate around 4800 operations per second. Since all the operations in the circuit are sequential, this is also the speed at which each step of the numerical integration is performed. Assuming that you need around 10–20 steps to reach the stable point, we could be processing more than 200 points per second. Then, the full resolution would be a matter of how many parallel threads can be implemented into an FPGA or an ASIC, but the numbers show that it seems to be adequate for real-time processing, even using low-speed systems.

Notice that in this paper only the proof of concept for the SC Memristive CNN has been discussed, not comparing it against any other improving algorithm using, for instance, a hard-wired algorithm implementation or Neural Networks, which may show much better image improvement. It has to be noted, however, that the method presented here is training-free, which simplifies the design when compared against NNs and also removes any possible bias introduced by the training. In addition, the facility to change the algorithm is also worth mentioning, since it reduces to changing the values of the coefficients in the the cells. This makes this approach especially preferable over a hard implementation of specific algorithms in, for instance, multi-purpose systems that can need to be swapping functions on the fly.

## Figures and Tables

**Figure 1 micromachines-13-00067-f001:**
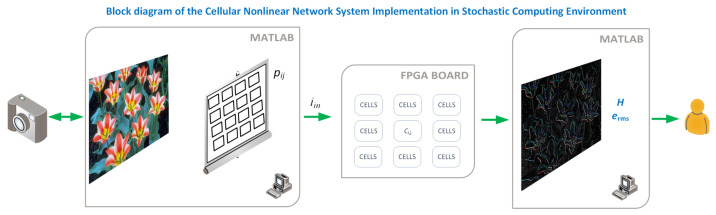
Conceptual depiction of the system, showing the tasks assigned to Matlab and those performed by the FPGA. The FPGA and Matlab are used jointly by using the FPGA-in-the-loop tool from Matlab, where the VHDL code is automatically generated, uploaded, and integrated with the main script at the computer.

**Figure 2 micromachines-13-00067-f002:**
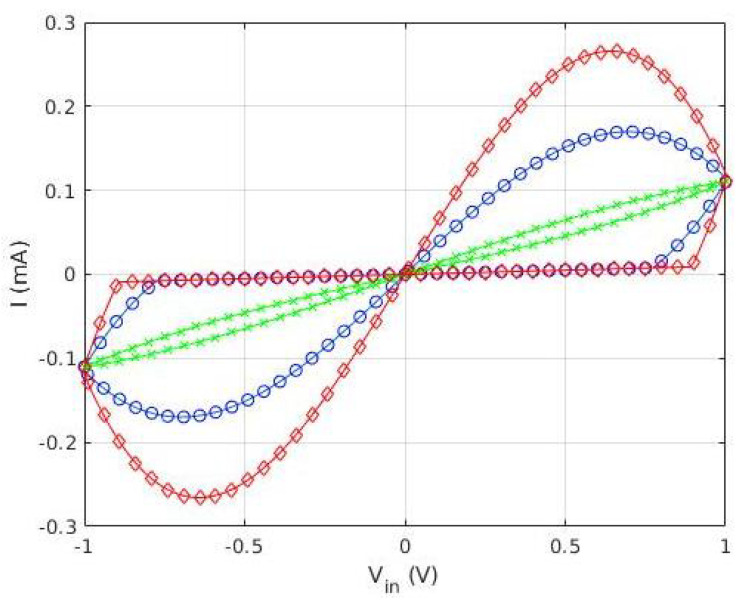
Representation of the I-V characteristics of the memristor defined by Equations (Equation 1) and (Equation 6) for $G_{0}=0.1\;\mathrm{mS}$, $\phi_{0}=10$, and three different frequencies (ω(red ⋄) < ω(blue o) < ω(green x)).

**Figure 3 micromachines-13-00067-f003:**
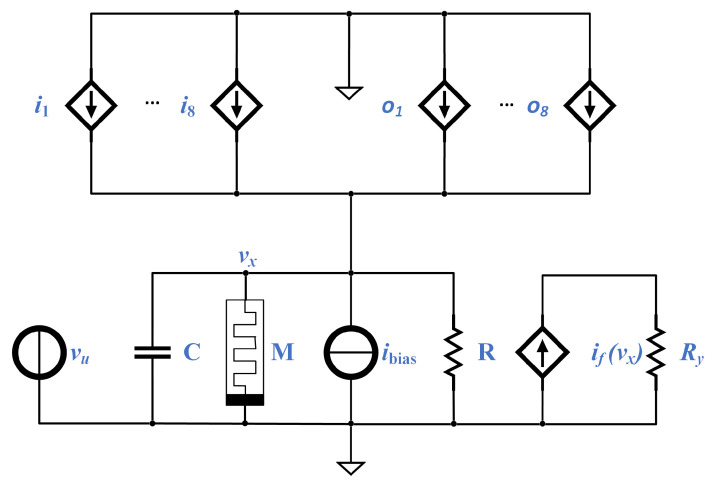
Schematics and Stochastic implementation of a CNN cell, the processing element in cell C(i,j) (i∈{1, ⋯, M}, j∈{1, ⋯, N} ). The other elements (resistor, memristor, and capacitor) present the same values from cell to cell, i.e., $C_{xi,j}=C_{x}$, $m_{xi,j}=m_{x}$, and $R_{yi,j}=R_{y}$. Adapted from [3].

**Figure 4 micromachines-13-00067-f004:**
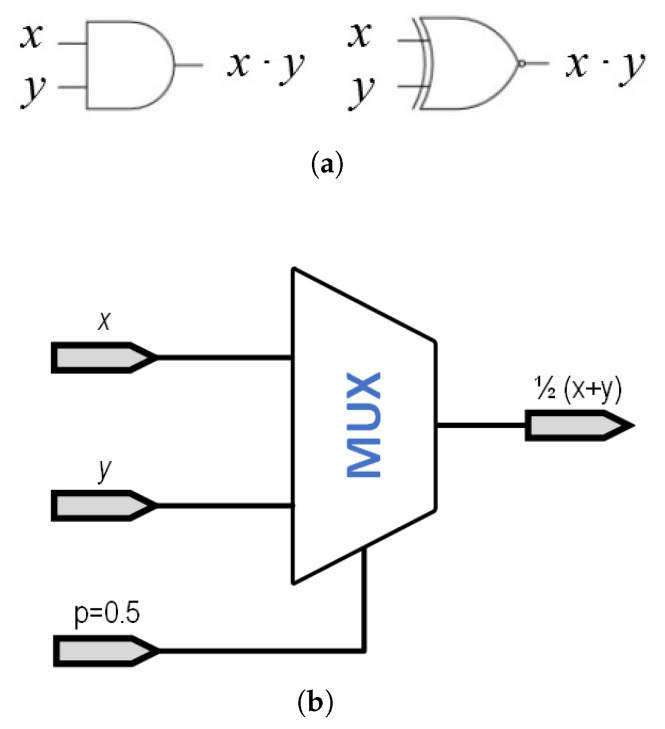
Basic implementation of basic operation in SC. (**a**) Basic implementation scheme of a SC multiplier in the (0⋯1) range (AND gate, left) and in the (−1⋯1) range (XNOR gate, right). (**b**) Basic implementation scheme of a SC adder using a multiplexer.

**Figure 5 micromachines-13-00067-f005:**
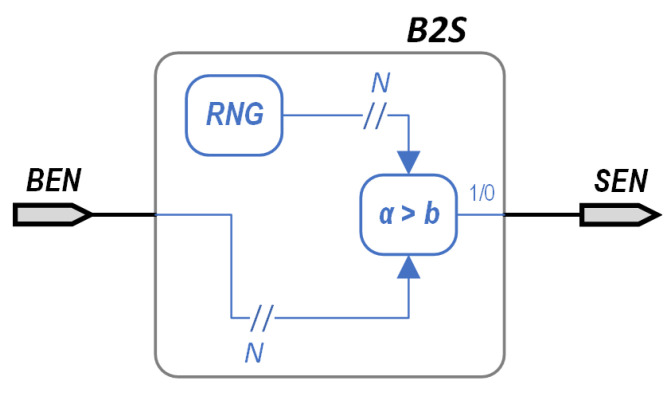
Basic implementation scheme of a Binary Encoded Number (BEN) to a Stochastic Encoded Number (SEN), using a Random Number Generator (RNG).

**Figure 6 micromachines-13-00067-f006:**
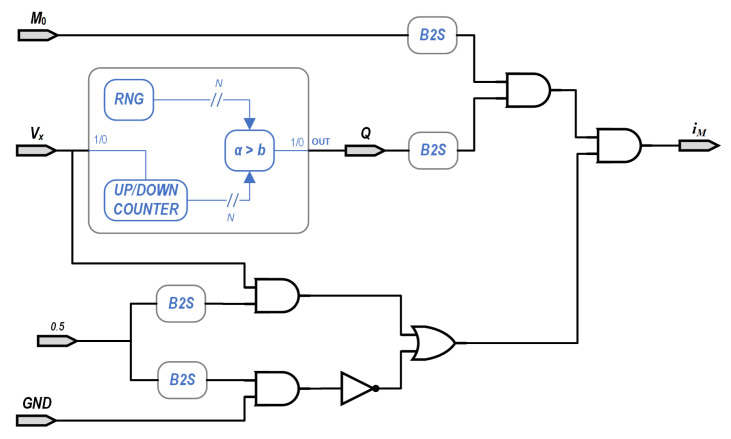
The Stochastic Computing memristor implementation. Inputs $v_{x}$ and *GND* are the SCN values of the positive and negative inputs of the memristor, respectively, and $i_{M}$ is the calculated SCN value of the current.

**Figure 7 micromachines-13-00067-f007:**
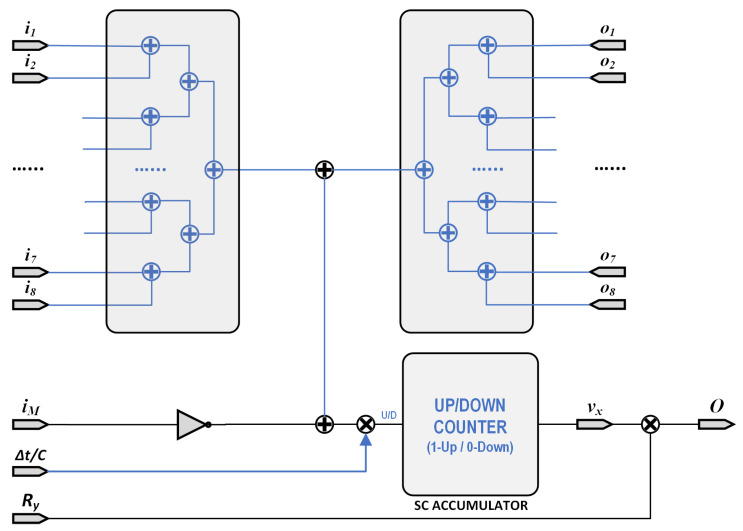
Stochastic Computing Circuit implementation of the M-CNN processing element C(i,j) as in Figure 3.

**Figure 8 micromachines-13-00067-f008:**
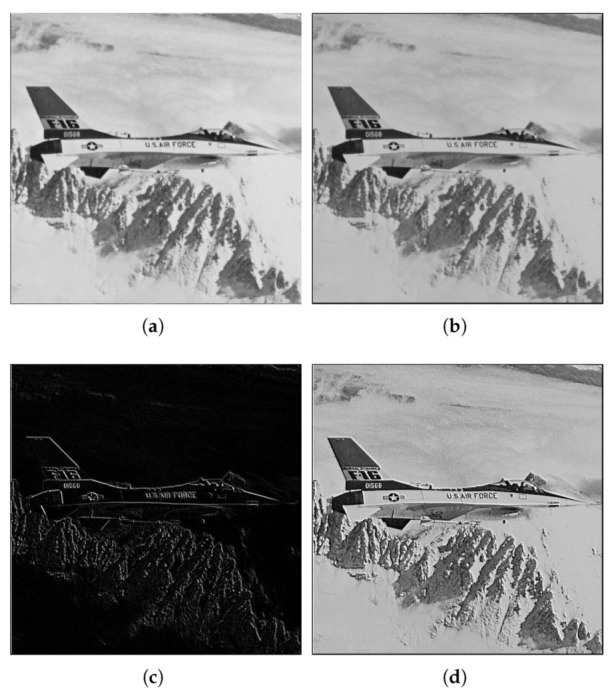
Example 1: results obtained using the three proposed stochastic computing CNN with different gene values. (**a**) Original image. (**b**) Stored image. (**c**) Edge detection result. (**d**) Sharpened image.

**Figure 9 micromachines-13-00067-f009:**
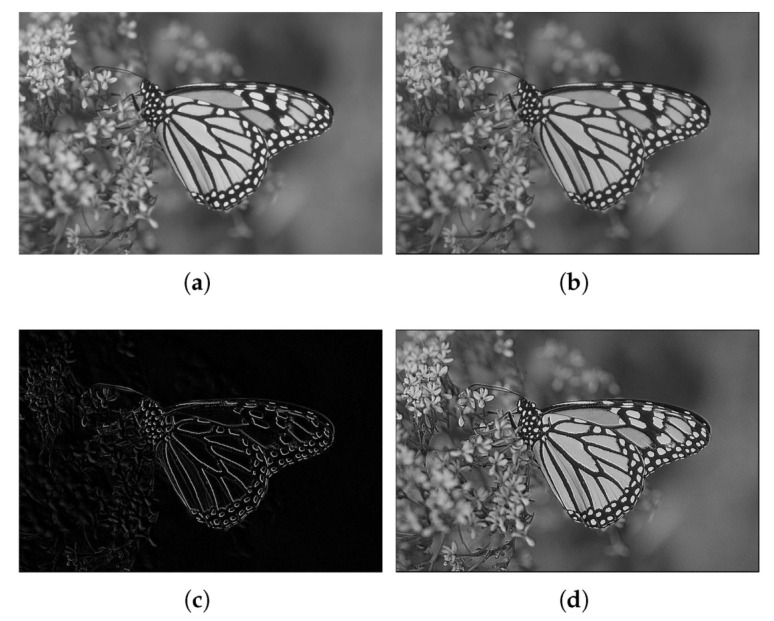
Example 2: results obtained using the three proposed stochastic computing CNN with different gene values. (**a**) Original image. (**b**) Stored image. (**c**) Edge detection result. (**d**) Sharpened image.

**Figure 10 micromachines-13-00067-f010:**
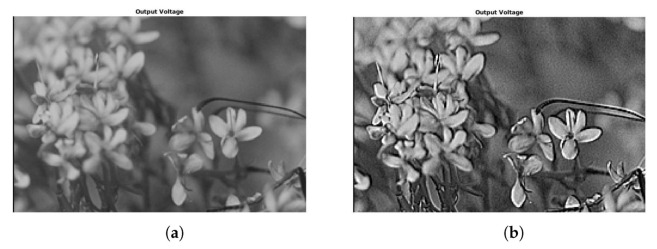
Example 2: Zoom-in of Figure 9a,d, showing a detail of the sharpening results obtained using the proposed stochastic computing CNN. (**a**) Zoom-in of the original image. (**b**) Zoom-in of the sharpened image.

**Figure 11 micromachines-13-00067-f011:**
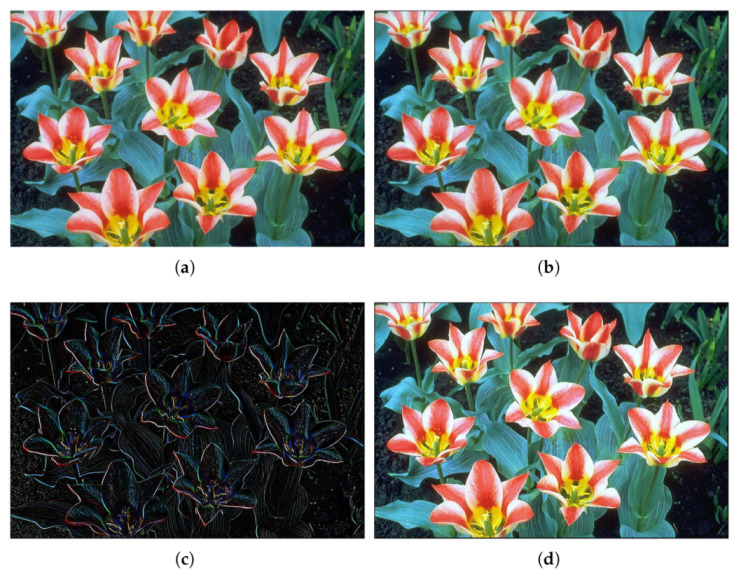
Example 3: Color figure, showing the sharpening results obtained using the proposed stochastic computing CNN. (**a**) Original image. (**b**) Stored image. (**c**) Edge detection result. (**d**) Sharpened image.

**Figure 12 micromachines-13-00067-f012:**
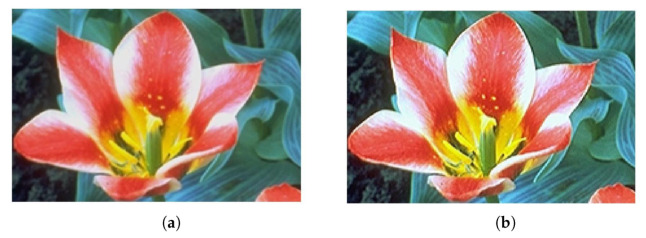
Example 3: Zoom-in of Figure 11a,d, showing a detail of the sharpening results obtained using the proposed stochastic computing CNN. (**a**) Zoom-in of the original image. (**b**) Zoom-in of the sharpened image.

**Table 1 micromachines-13-00067-t001:** Parameter values for the elements of the circuit in Figure 3, where the memristor is defined by Equation (Equation 1).

Parameter	Value
R	100 kΩ
C	50 μF
$G_{0}$	100 kΩ
$\phi_{0}$	10 V·s

**Table 2 micromachines-13-00067-t002:** Coefficient notation for M=A,B(m=a,b). Notice that the coefficient for the current node is (0,0).

*M*	$\left(\begin{array}{ccc} m_{-1,-1} & m_{0,-1} & m_{1,-1}\\ m_{-1,0} & m_{0,0} & m_{1,0}\\ m_{-1,1} & m_{0,1} & m_{1,1} \end{array}\right)$

**Table 3 micromachines-13-00067-t003:** Picture size in pixels. The color depth is 8 bits per channel.

Figure	Size (M × N ) Pixels${}^{2}$
Figure 8	512×512
Figure 9	768×512
Figure 11	768×512

**Table 4 micromachines-13-00067-t004:** Calculated values of entropy of the images. The results for the color image show the three color planes separately.

Figure	Original	Store	Edge	Enhance
Figure 8	6.70	6.71	4.03	6.90
Figure 9	7.18	7.20	3.87	7.45
Figure 11 (R)	7.15	7.07	4.04	7.38
Figure 11 (G)	7.16	7.27	4.19	7.88
Figure 11 (B)	7.16	7.28	4.26	7.76

**Table 5 micromachines-13-00067-t005:** Coefficients for the input and output weights in Equation (Equation 24) for the case of the image store setup.

*A*	$\left(\begin{array}{ccc} \;0.0\;\; & \;0.0\;\; & \;0.0\;\;\\ \;0.0\;\; & \;0.9\;\; & \;0.0\;\;\\ \;0.0\;\; & \;0.0\;\; & \;0.0\;\; \end{array}\right)$
*B*	$\left(\begin{array}{ccc} \;0.0\;\; & \;0.0\;\; & \;0.0\;\;\\ \;0.0\;\; & \;0.1\;\; & \;0.0\;\;\\ \;0.0\;\; & \;0.0\;\; & \;0.0\;\; \end{array}\right)$

**Table 6 micromachines-13-00067-t006:** Calculated RMS of the stored images, referred to the original image. The results for the color image show the three color planes separately.

Figure	RMS
Figure 8b	1.29%
Figure 9b	0.57%
Figure 11b (R)	1.13%
Figure 11b (G)	0.58%
Figure 11b (B)	0.61%

**Table 7 micromachines-13-00067-t007:** Coefficients for the input and output weights in Equation (Equation 24) for the case of the edge detection setup.

*A*	$\left(\begin{array}{ccc} \;0.0\;\; & \;0.0\;\; & \;0.0\;\;\\ \;0.0\;\; & \;0.0\;\; & \;0.0\;\;\\ \;0.0\;\; & \;0.0\;\; & \;0.0\;\; \end{array}\right)$
*B*	−1/8−1/8−1/8−1/8−1.0−1/8−1/8−1/8−1/8

**Table 8 micromachines-13-00067-t008:** Coefficients for the input and output weights in Equation (Equation 24) for the case of the image enhancement setup.

*A*	$\left(\begin{array}{ccc} \;0.0\;\; & \;0.0\;\; & \;0.0\;\;\\ \;0.0\;\; & \;0.3\;\; & \;0.0\;\;\\ \;0.0\;\; & \;0.0\;\; & \;0.0\;\; \end{array}\right)$
*B*	−0.05−0.05−0.05−0.05−0.43−0.05−0.05−0.05−0.05

## Data Availability

All the images used in the work are publicly available in the internet for free download.

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
