# Peer review of "Stochastic Computing Emulation of Memristor Cellular Nonlinear Networks"

_micromachines, 2021, doi:10.3390/mi13010067_

Round 1
Reviewer 1 Report
The manuscripts by Camps et al. reported using stochastic computing in design and implementing cellular nonlinear networks, which is prototyped on FPGA. The results are interesting and technically sound. The following comments shall be addressed before publication.
- The cellular neural network is based on the memristor proposed in Eqs. (3)-(6), which is a specific example for the suggestion in Eq. (3). What’s the evidence for selecting parameters (i.e., G0, φ0) in Eq. (6)?
- Please clarify the meaning of zij and add related description.
- There are many references reported that the memristor-based cellular neural network was used to image processing, i.e., edge detection, image inversion, etc [IEEE TRANSACTIONS ON NEURAL NETWORKS AND LEARNING SYSTEMS, vol. 26, no. 6, pp. 1202-1213, 2015]. What’s the advantage of this work? Does it have some differences using this kind of memristor?
Author Response
Q: The cellular neural network is based on the memristor proposed in Eqs. (3)-(6), which is a specific example for the suggestion in Eq. (3). What’s the evidence for selecting parameters (i.e., G0, φ0) in Eq. (6)?
Answer: Parameter values have been chosen to be similar to those proposed in [37], to optimize the emulator behavior. Another option would have been using the process described in [3,32] or [38], but then the scales of the values were too different to allow the emulator to perform efficiently. We have added a comment on this at the beginning of section 4, close to table 1 (it appears in red).
Q: Please clarify the meaning of zij and add related description.
Answer: We thank the reviewer for pointing out this. The variable zij is a constant used to define the stability of the system. It is equivalent to the ibias later on in the paper. We have added these comments on the paper in red.
Q: There are many references reported that the memristor-based cellular neural network was used to image processing, i.e., edge detection, image inversion, etc [IEEE TRANSACTIONS ON NEURAL NETWORKS AND LEARNING SYSTEMS, vol. 26, no. 6, pp. 1202-1213, 2015]. What’s the advantage of this work? Does it have some differences using this kind of memristor?
Answer: We have added the citation to Duan et al. Related to the question of the advantage of this work, it is, up to our knowledge, the first time when an actual hardware implementation has been proposed using memristors emulators.
Reviewer 2 Report
This paper deals with memristive cellular nonlinear networks profiting from the stochastic computing paradigm. Mainly, the authors implemented the CNN structure with the help of FPGA and received some good results in terms of efficiency. However, I am not sure if the idea is totally practical for adequate real-time application needs. My comments are as follows.
Please discuss the applicability of your proposal with image processing speed and whether it is adequate for real-time needs.
More related works and some recent state-of-the-art techniques are required for the paper.
Compare some of the image processing results with other works, if possible. The paper also needs further analysis tests and results.
Also, please mention datasets and any relevant details of FPGA implementation.
What are the chances of failure outcomes with the FPGA implementation? The authors may need to discuss some parts of the limitations of the FPGA implementation accordingly.
Author Response
Question: Please discuss the applicability of your proposal with image processing speed and whether it is adequate for real-time needs.
Answer: We thank the reviewer for the comment. As stated in the text, we are using 14-bits stochastic numbers, which translates into a chain length of 2^14=16384. A very entry-level FPGA can run at 80MHz, so it could operate around 4800 operations per second. Since all the operations in the circuit are sequential, this is also the speed at which each step of the numerical integration is performed. Assuming that you need around 10-20 steps to reach the stable point, we could be processing more than 200 points per second. Then, it’s a matter of how many parallel threads can be implemented in an FPGA or an ASIC, but the numbers show that it seems to be adequate for real-time processing, even using low-speed systems. We have added some comments along this line in the text.
Question: More related works and some recent state-of-the-art techniques are required for the paper.
Answer: We kindly remind the reviewer that the aim of the paper is showing the feasibility of implementing a CNN utilizing memristors and furthermore achieve this within a stochastic computing environment, and not proposing new image processing techniques. Toward this goal, we have mentioned some of the most relevant (in our opinion) and recent papers on stochastic computing and CNN design. Then, we have used the implemented CNN to perform some image processing tasks, as in [39] for providing some representative use case examples.
Question: Compare some of the image processing results with other works, if possible. The paper also needs further analysis tests and results.
Answer: In our opinion it is not feasible to include any comparison of the presented in the paper stochastic implementation of Memristor Cellular Nonlinear Networks in hardware, to other similar implementations, since the latter are implemented in software. Additionally, as mentioned in the title, the abstract and the introduction the main goal of this paper is not the image processing but the successful hardware implementation of a memristor-based CNN within a stochastic computing environment. Finally, we believe that the results presented are adequate enough to support this successful implementation of the specific CNN.
Question: Also, please mention datasets and any relevant details of FPGA implementation.
Answer: We have added a mention on the web page where we have downloaded the pictures from (http://www.hpca.ual.es/~vruiz/images). We have also added a mention on the implementation of the RNG algorithm (a Mersenne twister version). All the rest of the FPGA implementation is standard, and can be implemented readily from the paper, since there are only ORs, ANDs, multiplexers and accumulators.
Question: What are the chances of failure outcomes with the FPGA implementation? The authors may need to discuss some parts of the limitations of the FPGA implementation accordingly.
Answer: We don’t really understand the meaning of the reviewer about “failure outcomes”. If it refers to hardware failures, as in a broken module, in Duan et al [39] they show that CNN are resilient even in the case of a “pixel” failure (we have added a comment on this in the conclusions). If the reviewer is referring to a “mathematical” error (overflow, etc), the only reasonable source of this kind of problem could be the accumulator, which is limited both at the maximum value (2^14) and at the minimum (0), as stated by Eq 20 and 21. This is actually equivalent to a memristor, where there are maximum and minimum conductance values.
Round 2
Reviewer 1 Report
I thank the authors for the prompt responses which have well addressed my early concerns. I thus recommend it for publication as is.